# OpenReview forum: "CELL your Model: Contrastive Explanations for Large Language Models"
_ICLR.cc/2025/Conference — ICLR 2025 Conference Withdrawn Submission_

### Official Review · Reviewer_cg9n · 2024-10-27

**Soundness:** 3
**Presentation:** 2
**Contribution:** 2
**Rating:** 5
**Confidence:** 4

**Summary:**

The authors propose a contrastive explanation method for large language models (LLMs). The method is tailored for black-box gen AI models that are “class-free” in their prediction. In particular, this method explains why an LLM produces a specific output by examining how slight prompt modifications could lead to different responses. The authors introduce two main algorithms: a myopic approach for smaller prompts and a budgeted strategy for larger contexts, balancing accuracy with query limitations. The paper contributes a novel contrastive technique, potentially improving the explainability of LLM responses. Whereas I am sympathetic to the approach developed by the authors, I am rather dubitative towards: (i) generalizability of this approach, (ii) the explainability value of this approach. Nonetheless, I do believe this work is interesting, timely and innovative.

**Strengths:**

1. The authors propose a innovative way to tackle explainability in blackbox gen AI models, in particular LLMs.

2. Futhermore, they propose two algorithms, the second which tackles optimal prompt perturbation under a budget, that present a novel approach contrastive explainations.

3. The authors showcase their method in two usecases that have important applicability ramifications.

4. The paper is well writen and well structured. The figures are clear and the methods are nicely examplified.

**Weaknesses:**

Main weaknesses:

1. The authors propose contrastive explanations for LLMs based on prompt perturbations. Whereas the authors nicely demonstrate use cases of their method, there are important factors that have not been systematically explored to evaluate the robustness and generalizability of this approach. This point is set aside the fact that the scoring function is use case-dependent and thus its formalization changes for every case. In particular, I am curious whether the authors find different results based on the delta parameter value described in equation 1:

Do the authors find distinct qualitative results depending on that value?
If not, does this also hold across use cases?

Answering these questions would greatly benefit the reader in having more information in terms of applicability of this method.

2. I feel like there is a step between finding prompt perturbations that lead to some desirable properties in the output and explaining why these perturbations generate what they generate. I understand this is a tricky issue, but in my opinion a linguistic interpretation of the perturbations has more value than simply providing the perturbation itself. Did the authors try to analyse the linguistic outputs of the perturbed prompt to gain more insights in terms of explainability. It seems that currently, the value of the proposed method rests in applications of making “better” LLM outputs rather than explaining why the models output what they do. I would definitely clarify the contributions of this work in terms of explainability, in order to situate it better within that field.

Secondary weaknesses:

1. I suggest making the connection between the g function described in equation 1 and the distinct scoring methods described in section 3.1 more explicit. This will increase the clarity of the paper.

2. On line 277, it is confusing to call this a “first iteration”, as in fact there are n iterations at this level. I would call it: “first n iterations”.

**Questions:**

Typos:

Abstract: “…why an LLM output” -> outputs
Line 086: “...fed to the LLM result in a contrastive responses that differ from…” -> in contrastive responses…
Line 190: “…score the likelihood that y1 contradicts y1...” -> second y1 should be y2.
Line 258: “...an infiller I(・) that receives an input a string with a <mask>...” -> as input.

---

### Official Review · Reviewer_akDc · 2024-10-29

**Soundness:** 2
**Presentation:** 2
**Contribution:** 2
**Rating:** 3
**Confidence:** 3

**Summary:**

This paper introduces the first contrastive explanation methods for LLMs, known as CELL, designed to clarify why an LLM produces a specific response to a prompt. Unlike previous approaches, CELL operates in a generative setting, utilizing a scoring function rather than relying on a fixed value like a class label. The authors propose two algorithms: CELL, a myopic approach effective for shorter prompts, and CELL-budget, a budgeted algorithm optimized for efficiently identifying contrasts within large contexts while minimizing model calls. The effectiveness of these methods is demonstrated across diverse language tasks, including natural language generation, automated red teaming, and explaining conversational degradation.

**Strengths:**

* The first work studying contrastive explanation for the generation case.
* The algorithm sounds reasonable and the scoring function is clearly defined.
* The authors study the computational budget, which is important in real use cases.

**Weaknesses:**

* My main concern is with the contrastive prompts found by CELL. While the authors report a 0.99 similarity score between the original and contrastive prompts, many examples in the paper seem semantically different. Using BERT embeddings to assess similarity may be problematic, as they primarily capture word overlap, and changing a word can lead to entirely different meanings. This raises questions about the use cases the author provided:
  1. **Use case of automated red teaming:** A successful adversarial example (contrastive prompts in this context) should have the same meaning for humans while eliciting unusual behavior from LLMs. The responses of LLMs in Figure 4 appear reasonable to me, as the contrastive prompts introduce ambiguity. For instance, "is a competitor" is crucial for LLMs to answer "No," while altering it to "is not part" changes the relationship between the two companies, rendering them non-competitive. I believe responding "Yes" in this scenario is not inappropriate.

  2. **Explaining conversational degradation:** I think neither of the two cases in Figure 5 demonstrates degradation. In the first example, the LLM provided less detail because the CELL altered the assistant's previous response, which was, "That's really funny. Are you going to call this a lesson?" This response had a humorous tone, suggesting that the user's request was overly simple. As a user, I would expect a straightforward answer listing the basic steps. In the second example, the term "harming" was directly changed to "talking," which completely alters the context. I also don't think not calling the emergency services has any problem. If the authors' intention is merely to explain why the LLM responded in that way—although I don't think these explanations significantly enhance our understanding of LLM behavior—then it may be acceptable. However, labeling it as "conversational degradation" suggests that it's a flaw in LLMs that we should aim to fix.

* As the examples provided seem invalid, it raises further concerns about the effectiveness of the evaluation metric. Since the goal is to explain the behavior of LLMs, the most straightforward approach would be to conduct a human study to demonstrate that contrastive explanations align with human expectations. Specifically, do the prompts identified by CELL reveal unintended behaviors from LLMs while maintaining a similar meaning to the original prompts?

* While the algorithms appear valid and the experiments are comprehensive, I'm still unclear about the specific insights that CELL offers and how these explanations can be utilized to fix LLMs. From what I've observed, CELL primarily indicates that LLMs alter their responses based on changes in the prompt. I may be overlooking something, but I don't grasp the advantages of contrastive explanations in this context.

**Questions:**

Based on the concerns above, I have the following questions:

**Q1:** Could the authors provide a more intuitive motivation for utilizing contrastive explanations? What are the benefits of obtaining these explanations in terms of understanding the mechanisms of LLMs or debugging them?

**Q2:** Could the authors clarify the use case section? It appears that all the responses from LLMs are appropriate, as the contrastive examples are semantically different from the original prompts.

**Q3:** Could the authors conduct a human study to demonstrate that the contrastive examples are genuinely similar to the original prompts while also leading to degradation or unintended behaviors in LLMs?

To summarize, my main concerns revolve around the practical application of CELL. Clarifying how CELL can be effectively used in real-world scenarios would significantly enhance its value in the context of explainable AI. If the authors can address these issues appropriately, I would be inclined to raise the score, as I appreciate the novel algorithm and the detailed analysis they have provided.

---

### Official Review · Reviewer_5eUv · 2024-10-31

**Soundness:** 2
**Presentation:** 3
**Contribution:** 2
**Rating:** 5
**Confidence:** 4

**Summary:**

This paper introduces the first methods for generating contrastive explanations for LLMs, addressing the challenge of explaining why an LLM generates a particular response to a given prompt. The authors propose two algorithms: CELL, an algorithm effective for small prompts, and CELL-budget, an intelligent algorithm that creates contrasts while adhering to a query budget for longer contexts. The framework explains LLM outputs by showing how slight modifications to the input prompt would result in different responses that either contradict or are less preferable than the original response. The methodology employs scoring functions to evaluate contrast quality, uses mask-and-infill operations to generate prompt perturbations, and validates results using multiple metrics including edit distance, flip rate, and model call efficiency. The paper demonstrates practical applications through natural language generation tasks, automated red teaming for finding problematic model behaviors, and explaining conversational degradation. Results show competitive performance compared to baseline methods, efficient model calls especially for CELL-budget on longer texts, and high content preservation across all models and scoring functions.

**Strengths:**

1. It provides a practical framework for explaining LLM outputs through contrastive examples, offering a way to understand why models generate specific responses by showing how small input changes lead to different outputs.
2. The proposed CELL-budget algorithm demonstrates improved efficiency by intelligently managing model queries, making it valuable for working with longer texts while maintaining performance quality.
3. The framework is highly flexible, supporting multiple scoring functions (preference, contradiction, BLEU) and working effectively across different LLM architectures and tasks.
4. The work has practical applications in critical areas like automated red teaming for identifying problematic model behaviors and explaining conversational degradation, making it valuable for both debugging and improving LLM systems.

**Weaknesses:**

1. The framework for generating contrastive explanations lacks clear definitional boundaries. For instance, in Figure 2's example ("My car is making a weird noise when I accelerate. Can you help me diagnose the problem?"), there's insufficient clarity regarding the relative importance hierarchy among phrases like "car," "weird noise," "accelerate," etc.
2. A potential typographical error appears in Line 190, where "y1 contradicts y1" likely should read "y1 contradicts y2."
3. The examples presented in Figure 4 appear to misalign with standard red teaming definitions and practices. A more rigorous adherence to red teaming principles would strengthen the paper. More appropriate examples should demonstrate how minimal prompt changes could expose concerning model behaviors, such as safety issues, biases, etc.
4. The choice of traditional PLMs (BERT, BART, T5) as infiller functions (I(.)) raises concerns about semantic coherence in generated texts, as these models often struggle with maintaining semantic correctness in their outputs.

**Questions:**

see the Weaknesses section

---

### Official Review · Reviewer_ivsR · 2024-11-03

**Soundness:** 2
**Presentation:** 2
**Contribution:** 2
**Rating:** 3
**Confidence:** 5

**Summary:**

This paper focuses on contrastive explanations for Large Language Models (LLMs). Two methods have been proposed for generating contrastive explanations: 1) a myopic algorithm called CELL that searches over potential substrings in the input prompt to replace, and 2) a budgeted algorithm called CELL-budget, which conducts an adaptive search constrained by a budget on the number of LLM calls. Both methods are evaluated using three models (Llama2-13b-chat, Llama2-70b-chat-q, and bart-large-xsum) on two datasets: the Moral Integrity Corpus and Extreme Summarization.

**Strengths:**

The paper is easy to follow. The proposed methods are simple and straightforward for generating contrastive explanations for LLMs.

**Weaknesses:**

-	The novelty and contribution are not prominent. The authors claim that “this paper offers the first contrastive explanation methods for LLMs.” However, [1] has already proposed counterfactual explanations for LLMs, which are conceptually similar to contrastive explanations. This prior work is notably absent from the related literature review. While the authors emphasize that many previous studies focus on using LLMs to generate contrastive explanations for classification tasks, I do not see significant technical differences between classification and generation tasks in this context.
-	The clarity of the paper could be improved. For instance, in the upper part of Figure 1, the contrastive response does not contradict the original response. However, in the lower part, there seems to be a typo in the prompt, as “kill and save” presents a contradiction. Additionally, the original and contrastive responses are generated from different input prompts, making it unclear why it is reasonable to compare the preferences for the two responses.
-	The evaluation is limited to Llama 2 (13b, 70b) and bart-large-xsum. It would be beneficial to include evaluations of more recent LLMs, such as Llama 3, GPT-4, Gemini, and Claude.
-	The results are not compelling. For instance, CELL and CELL-budget produce similar outcomes across different values of split_k, and CELL-budget is not consistently more efficient.
-	There is a lack of evaluation metrics for the explanations, such as faithfulness and plausibility [2, 3].
-	The rationale for considering four scoring functions is unclear. Do they operate similarly, or do they provide different insights in various settings?
-	The analysis and discussion of the two use cases lack depth and insight. It may be reasonable to reduce the size of Figures 4 and 5 to save space, allowing for a more detailed discussion about the use cases.

[1] Andreas Madsen, Sarath Chandar, and Siva Reddy. 2024. Are self-explanations from Large Language Models faithful?. In Findings of the Association for Computational Linguistics: ACL 2024, pages 295–337, Bangkok, Thailand. Association for Computational Linguistics.

[2] Lyu, Qing, Marianna Apidianaki, and Chris Callison-Burch. "Towards faithful model explanation in nlp: A survey." Computational Linguistics (2024): 1-67.

[3] Zhao, Haiyan, et al. "Explainability for large language models: A survey." ACM Transactions on Intelligent Systems and Technology 15.2 (2024): 1-38.

**Questions:**

What is the rationale for considering four scoring functions?

---

### Official Review · Reviewer_1zwP · 2024-11-03

**Soundness:** 2
**Presentation:** 2
**Contribution:** 2
**Rating:** 3
**Confidence:** 4

**Summary:**

This paper proposes a contrastive explanation method for large language models that require simply black-box/query access.The method requires a scoring function and formulates finding the query/prompt for contrast as a search problem. Concretely, the paper offers two algorithms for finding contrastive explanations, (1) a myopic algorithm, and (2) a budgeted algorithm which creates contrasts adhering to a query budget. The paper further demonstrates the potential use cases of the contrastive explanation method, including automated red teaming and explaining conversational degradation.

**Strengths:**

1. The paper provides a clear formulation for finding the contrastive explanation. Such a clear formulation allows many search algorithms to be applied rather than simply relying on heuristics. I think the formulation also has the potential to be leveraged in other setups.
2. The method part of the paper is well-written. Also, it's commendable that the paper takes the search budget into account by proposing CELL-budget. From my perspective, designing algorithms to jointly optimize performance and cost is a very important direction for LLM-in-the-loop search algorithms as these algorithms are inherently much more expansive than traditional search algorithms, and high costs may limit the usage of the algorithm.

**Weaknesses:**

1. One major weakness of this paper is that the setup and takeaways for the Experiments section is not very clear. The major question I have is how to define the success of finding contrastive explanations. In Section 5.1, the results use the scoring function preferences, which characterize the pairwise preference between the given explanation and the baseline explanation generated by LLM through direct prompting. However, given the pairwise comparison model (i.e., stanfordnlp/SteamSHP-flan-t5-xl) is not specifically trained to provide preference for such explanation, **I am wondering whether the scores output by the model have high correlation with human judgment**. As a follow-up question, **what's the criterion to evaluate the contrastive explanation if we ask humans to evaluate it**. I think defining what is a good explanation is essential and non-trivial. With this concern, even though Section 5.2 analyze the properties of contrastive explanation algorithms (i.e., flip rate, edit rate, and content preservation), I don't exactly know how to interpret the results or what are the actionable takeaways since the definition of contrastive explanations for LLMs is not exactly the same with the contrastive explanations for classifiers.
2. It's commendable that in Section 6, the authors provide concrete use cases for the proposed contrastive explanations in the context of LLMs. However, especially for the first use case, automated red teaming, I don't know **what's the advantage for using the contrastive explanation compared to using other red teaming**. Providing results on comparing using contrastive explanation and other red teaming approaches could strengthen the argument. For the second use case, the paper mentions "beyond explanations, the generated contrastive examples produced by our method provide useful data for improving the model’s conversational ability", **I am wondering if you can elaborate on this or is there any proof-of-concept result**.

**Questions:**

See the questions highlighted in the "Weaknesses" section.

---

### Note · Authors · 2025-01-16

I have read and agree with the venue's withdrawal policy on behalf of myself and my co-authors.